# Least Privilege Access for Persistent Storage Mechanisms in Web Browsers

## ABSTRACT

Web applications often include third-party content and scripts to personalize a user's online experience. These scripts have unrestricted access to a user's private data stored in the browser's persistent storage like cookies, localstorage and IndexedDB, associated with the host page. Various mechanisms have been implemented to restrict access to these storage objects, e.g., content security policy, the HttpOnly attribute with cookies, etc. However, the existing mechanisms provide an all-or-none access and do not work in scenarios where web applications need to allow controlled access to cookies and localstorage objects by third-party scripts. If some of these scripts behave maliciously, they can easily access and modify private user information that are stored in the browser objects.

The goal of our work is to design a mechanism to enforce fine-grained control of persistent storage objects. We perform an empirical study of persistent storage access by third-party scripts on Tranco's top 10,000 websites and find that 89.84% of all cookie accesses, 90.98% of all localstorage accesses and 72.49% of IndexedDB accesses are done by third-party scripts. Our approach enforces least privilege access for third-party scripts on these objects to ensure their security by attaching labels to the storage objects that specify which domains are allowed to read from and write to these objects. We implement our approach on the Firefox browser and show that it effectively blocks scripts from other domains, which are not allowed access based on these labels, from accessing the storage objects. We show that our enforcement results in some functionality breakage in websites with the default settings, which can be fixed by correctly labeling the storage objects used by the third-party scripts.

**ACM Reference Format:**
Anonymous Author(s). 2024. Least Privilege Access for Persistent Storage Mechanisms in Web Browsers. In *Proceedings of ACM Conference (Conference'17)*. ACM, New York, NY, USA, 10 pages. https://doi.org/10.1145/nnnnnnn.nnnnnnn

## 1 INTRODUCTION

Websites use persistent client-side storage mechanisms like cookies [15, 19], web storage (localstorage and sessionstorage) [10] and IndexedDB [17], as a means to store user- and site-specific data on the user's browser [25]. This provides websites the ability to maintain sessions and identify users over subsequent requests eliminating the need for users to authenticate themselves with the server

on every request. This also allows websites to persist other information across pages, e.g., e-commerce applications share cart and price details using cookies between the shopping and payment page.

These storage mechanisms are often used to store private and sensitive user information (e.g., session tokens); thus, their security is of paramount importance. *Third-party* scripts included on a web page have unrestricted access to this data stored by the host in the browser [16]. If some of these third-party scripts behave maliciously, they can access and alter the data stored in these persistent storage mechanisms. For instance, an adversary can get hold of an authentication cookie, and may use it to impersonate the user and initiate a session on behalf of the user. Similarly, if the adversary can replace the user's authentication cookie with their authentication cookie, the user would then perform actions on behalf of the attacker [30].

Browsers include security policies like the *same-origin policy* (SOP) [16] and *content security policy* (CSP) [44] to control the access of host resources by third-parties. However, SOP treats third-party scripts included on the host page as belonging to the same domain, thus providing access to all resources on that page, while CSP only controls the domains from which the scripts can be loaded on a page without specifying if/how each of these scripts can access the host's storage objects.

Attributes like the HttpOnly flag [22], Secure flag [6] and SameSite flag [43] were introduced to control the access of cookies by JavaScript (JS), sending cookies on unencrypted channels and on cross-site requests, respectively. Although the HttpOnly flag blocks *all* JS (including any third-party scripts) from accessing a cookie that has this flag set, various cases require cookies to be accessed by scripts and, in particular, by third-party scripts. For instance, analytics cookies are set by the host page and then accessed by third-party scripts such as Google Analytics to track user behaviors on websites; similarly, consent is generally managed by third-party consent management platforms (CMPs), which are included as third-party scripts that set and access the host cookies. Thus, completely blocking scripts from accessing all cookies is not practical in the real-world websites. The Secure and SameSite flags only control the inclusion of cookies sent along with HTTP requests, thereby allowing all scripts included on the page to access these cookies (sans the ones which have the HttpOnly flag set).

Objects stored in the web storage or IndexedDB are accessed via JS only and not included as part of HTTP requests. Unfortunately, no flags or attributes exist for providing an all-or-none protection mechanism and controlling the access of these objects by JS. These objects stored in the browser are freely accessible by any JS included on the host page without any restrictions making them vulnerable to confidentiality and integrity violations.

The goal of our work is to design a security mechanism that provides a fine-grained control of persistent storage objects by scripts included on a web page.

To realize this objective, we first analyze how extensively are storage objects accessed by third-party JS in real-world websites, and then present an approach to secure these accesses. We perform an empirical study of Tranco [34] top 10,000 websites that involved analyzing the access of persistent storage mechanisms by JS included on the web page. The use of these objects to store site-relevant data is very common as we show later in Section 4. We found that almost 95% of all cookies, 91% of all the localstorage objects and 74% of IndexedDB objects[1] in the browser are accessed by third-party scripts. Third-party accesses (to read or modify) are widely prevalent, and are 89.84%, 90.98% and 72.49% of all accesses in the case of cookies, localstorage and IndexedDB, respectively. Additionally, we found that in at least 16% of these third-party accesses, the third-party scripts read/modify cookies that are set by the host page (first-party); for localstorage, this access is around 10%. Section 4 discusses a more detailed analysis of these accesses.

Recent works [11, 12, 26, 35] discuss how some of these storage objects are being accessed by third-party scripts. However, they either focus on measurement and analyses of specific cookies used for tracking [35] or authentication and authorization [26], or on the use of web storage for tracking [11, 12]. Bahrami et al. [14] propose isolating cookies in the cookie jar based on the domains that set them, thereby preventing third-party scripts from accessing cookies created by the host page. However, as we show in Section 4, almost 45% of the cookies set by the host pages, in the top 10K websites, were read or modified by third-party scripts. Additionally, we observed that cookies created by third-party scripts are being accessed by the host page scripts (or first-party scripts) in ~5% of the cases and by other third-parties in almost 38% cases. Thus, a coarse-grained blocking of access to cookies set by another domain may result in functionality breakage on the host page. While some of these accesses may raise security and privacy concerns, we argue that the responsibility for granting or denying such access should be with the "owner" of these objects.

*Our approach.* We propose a fine-grained approach to control the reading and writing of storage objects by third-party scripts building on the *principle of least-privilege*. The central idea, described in Section 5, is to associate *labels* or *taints* with all storage objects to distinguish the objects set by the host page from the cookies set by third-party scripts. The labels are, then, used to determine whether a storage object is accessible by certain scripts or not, based on the attributes set by the "owners" of these objects.

To study the efficacy of our approach, we modify the Firefox web browser (Section 5) for carrying the context from JS to the DOM APIs that operate on the storage objects, and store labels (that are sets of domains) along with the objects. We enforce checks on third-party scripts accessing the storage objects by checking the labels on the objects against the scripts' domain, and evaluate this on 100 websites for functionality breakage. We show that with the default policy in place certain functionalities related to consent management and analytics do not run as expected. To ensure that these run correctly, the server needs to explicitly add the labels allowing access to third-party scripts.

To summarize, the key contributions of our work include: (1) a comprehensive empirical analysis of accesses to storage objects by

both first- and third-party scripts, (2) highlighting the limitations of the current browser policy providing third-party scripts an all-or-none access to storage objects, (3) the design and implementation of our least-privilege access approach and an evaluation of the impact of our intervention on website functionality breakage.

## 2 BACKGROUND

### 2.1 Persistent storage in browsers

Persistent storage objects are key-value pairs stored in the browser to maintain information across sessions or store site-relevant data. Cookies [15, 19] provide a mechanism for sharing state between clients and servers, which is useful in maintaining sessions. Servers authenticate users and maintain session information in the form of cookies in the users' browsers to identify them over subsequent requests; this eliminates the need for users to authenticate themselves with the server on every request. Cookies are also used to persist information across pages belonging to the same domain, e.g., e-commerce applications share cart and price details using cookies between the shopping page and the payment page. These cookies that are set by the host page are referred to as *first-party cookies*. Advertising scripts also use cookies to track a user's activity across different domains and display advertisements according to the user's behavior.

Web Storage API [10] and IndexedDB [17] were later introduced to address the size limitations of cookies. While web storage was originally introduced to store non-sensitive data like themes and languages [8], with the increase in the restrictions on cookies, developers have started using web storage to store sensitive information, as well, like cookie consent. IndexedDB is efficient when the size of data that needs to be stored is large. Although this makes integration of features easy, it opens another channel for information leaks to third-party scripts as these storage objects are shared between all scripts under the same origin.

The access of these storage objects is subject to certain policies, which we describe in the rest of this section.

### 2.2 Browser security policies

The same-origin policy (SOP) [16] is a standard that defines how documents and scripts of one *origin* are allowed to interact with resources belonging to another origin. An origin is identified using the protocol, hostname (or domain) and the port number, e.g., in `https://eg.com:443`, https is the protocol, eg.com is the domain name and 443 is the port number. SOP prevents resource access by frames embedded or included in a web page, if they belong to a different origin (e.g., `http://eg.com:80`), thus providing an all-or-none access control. However, SOP allows certain elements like images and scripts belonging to a different domain to be included in the context of the host page that loaded these elements.

*Content security policy* (CSP) [44] was later introduced for more fine-grained access control, wherein the server explicitly lists the domains from which various HTML elements or resources can be included on the web page, e.g., the following policy only allows scripts from example.com to be executed while any other dynamically loaded script shall not be executed:

```
Content-Security-Policy: script-src https://example.com/;
```

Although CSP allows more fine-grained policies to be specified, it

---

[1] An IndexedDB object is a key in a particular database

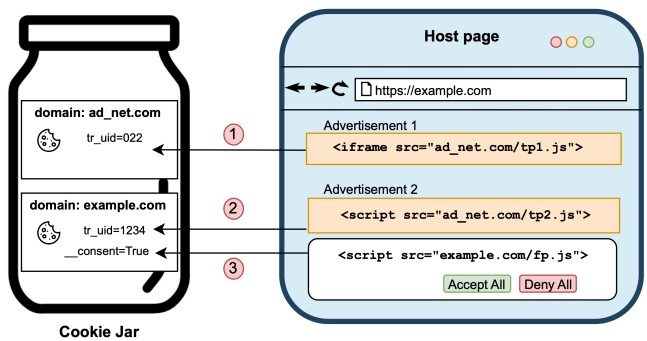

**Figure 1: Cookie access policies in Web browsers**

also allows an all-or-none access only, i.e., either all scripts from `https://example.com/` will be executed or none of them will. Thus, any third-party source that the host page may trust or include in the policy can execute on the host page and access any of the resources (e.g., cookies) without any restrictions.

## 2.3 Cookie access policies

Accessing cookies is subject to policies that are less restrictive than SOP, and subject to the values of `Domain`, `Secure` [6], `HttpOnly` [22] and `SameSite` [43] attributes. The `Domain` attribute defines which domain can access the cookie (irrespective of the protocol and the port number used); to only allow domains using the HTTPS protocol, the `Secure` attribute must be set for the cookie. However, as third-party scripts (originating from a domain other than the host page) included on the host page share the same origin as the host page, they have unrestricted access to the host's cookies in the browser. The `SameSite` attribute allows the developer to specify the context in which the cookie shall be shared over the network, which helps protect against cross-site request forgery (CSRF) [30] attacks. It does not, however, prevent a script from accessing or modifying the cookie value.

The `HttpOnly` flag is the only attribute that prevents *any* JavaScript code from accessing a cookie, when set to `True`. Even if the cookie needs to be accessed by a first-party script, for instance, when storing consent (Listing 1), the flag needs to be set to `False`, thereby restricting scripts belonging to the host domain from using these cookies, as well, when set to `True`.

One solution to prevent third-party scripts from accessing host cookies is to isolate them in iframes [16], but in many cases the website cannot function as intended unless they are included on the main page. For example, analytics libraries need to access specific user interaction metrics, such as mouse movements and clicks, which cannot be effectively captured if the script is executed within a separate iframe.

Figure 1 shows how scripts can access cookies in browsers. The host page `example.com` includes an advertisement script loaded in iframe from a third-party domain — `ad_net.com`. A cookie set by the script `tp1.js` (①) would be stored with `Domain=ad_net.com`. However, as the script `tp2.js` is included as a third-party script, the cookie set by the script (②) is stored with `Domain=example.com`;

`tp2.js` can also read or modify the cookie `__consent` set by the *first-party* script, `fp.js`, ③.

## 2.4 Web storage and IndexedDB access policies

Access to web storage objects happen similar to cookies except that web storage objects do not have any attributes other than the domain associated with them, i.e., any third-party script included on the host page can access all localstorage and sessionstorage objects set by the first-party scripts. The use of web storage as an alternative to cookies has increased significantly [11], thus allowing third-party scripts easy access to the data stored by the host page in the browser. Similarly, third-party scripts can access IndexedDB objects stored by the host page without any restrictions. While cookies have flags and attributes associated with them that provide them some protection, web storage and IndexedDB objects are not subject to any scrutiny other than the same-origin policy [16].

In the rest of the paper, we discuss examples, approaches and solutions using cookies, but these extend, without any loss of generality, to other persistent storage objects, as well.

## 3 MOTIVATING EXAMPLE

As there is no distinction between first-party and third-party scripts when accessing persistent storage, third-party scripts can easily access, modify or share storage objects set by the host domain. A naive solution to the problem of third-party scripts accessing persistent storage is to simply block all third-party JS from accessing any storage objects. While similar approaches have been proposed for third-party cookies [31], there are instances that require scripts to access storage objects.

An example that requires third-party scripts on web pages to access and modify cookies is when managing user consent (to comply with privacy regulations like GDPR [28, 29]). Web pages store users' privacy preferences and consent using storage objects. These objects are set and updated according to the user preference using JS, and hence cannot be simply blocked access to in scripts. Moreover, websites use third-party consent management platforms (CMPs) like TCF [27], OneTrust [4], TrustArc [42], etc. to maintain their users' privacy (as they are easy to integrate and maintain), each of which require access to first-party cookies.

However, unrestricted access can have negative implications. Consider, for instance, a website (Listing 1) that contains a consent banner, which provides an option to either accept or decline cookies. The user's consent decision is stored in `__consent`. The website also includes a third-party script to load an image on the page, which has access to all first-party cookies and storage objects. If this third-party script behaves maliciously (as shown in Listing 2), it can access and modify the `__consent` cookie as shown in Figure 2. The script may also modify the localstorage object providing incorrect analytics information about the user. As discussed in prior work [45], accessing first-party cookies can lead to confidentiality and integrity violations, the consequences of which include, but are not limited to, cross-site scripting (XSS), information leakage, cross-site request forgery (CSRF), and account hijacking.

*Cookie tossing:* A compromised or malicious script may also abuse this feature to perform a *cookie tossing* attack degrading

```
349  /* -------- example.com/fp.js -------- */
350  function showSelected(e) {
351      if (this.checked) {
352          document.querySelector('#output').innerText =
353                  'You selected to ${this.value} cookies';
354          if (this.value=='Decline') {
355              setCookie('__consent', "false", 30);
356          } else{
357              setCookie('__consent', "true", 30);
358      }}}
359  function noOfVisits() {
360    var cc = parseInt(localStorage.getItem("clickcount"));
361    localStorage.setItem("clickcount", ++cc);
362  }
```

**Listing 1: Cookie and localstorage access in first-party script**

```
366  /* -------- ad_net.com/bad.js -------- */
367  name = getCookie("__consent")
368  if (name != "")
369      if (name == "false")
370          setCookie("__consent", "true", 30);
371  var img = document.createElement("img");
372  img.src = "http://ads.bad?count="+
373          localStorage.getItem("clickcount");
```

**Listing 2: Malicious third-party script**

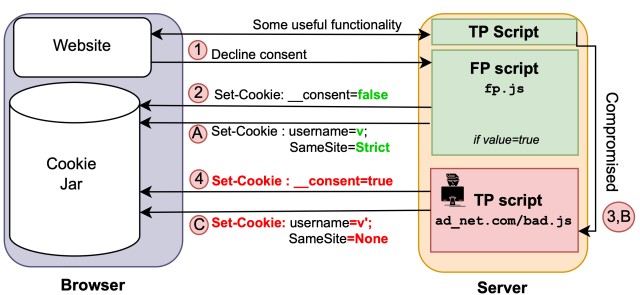

**Figure 2: Cookie tossing by malicious third-party script**

the security restrictions imposed on existing cookies. For instance, a cookie whose SameSite [43] attribute is set to "Strict" may be overwritten by a third-party script setting the SameSite attribute's value to "None". Figure 2 shows the workflow of a malicious third-party script changing host cookie's SameSite attribute to "None" (Ⓒ). Cookie tossing can affect cookies in three scenarios: (1) using cookies as authentication cookies. In this case, the cookies exploited are used for authentication and can effect websites which send authentication data through cookies. Some example scenarios include third-party SSO provider and online payments where sub-sessions are hijacked by injecting the attacker's account details; (2) associating important and session independent states with cookies result in attacker accessing the website with the user's account. This enables the attacker to access/hijack information about the user

like shopping history and browsing history. This scenario requires cookies to store session IDs in cookies which can be exploited by the attacker; (3) reflecting cookies into HTML involves injecting malicious script in reflecting cookies which launch an XSS attack in turn.

In this work, we propose an approach to ensure that none of the third-party scripts can read or modify cookies and other storage objects that they have not created, unless they are explicitly granted access to by their owner.

## 4 PREVALENCE OF STORAGE ACCESSES IN REAL-WORLD WEBSITES

To study the prevalence of third-party storage accesses in real-world websites, we performed a measurement analysis on Tranco top 10,000 websites[2]. We modified the Nightly Firefox version 98.0a1 (64-bit) browser to crawl these websites, and log the accesses of cookies, localstorage and IndexedDB objects by scripts included on the websites. This measurement study was performed in an automated manner running the Marionette test [9] during the months of March and April in 2024. For collecting the data, we instrumented the browser APIs — getCookie, setCookie, getItem, setItem, IDBObjectStore.get and IDBObjectStore.put — to capture all storage accesses in a log file. The logs capture the host's domain along with the domain of the origin of the script, extracted based on the URL from where the script was loaded. As all requests to read and modify storage objects are handled by these APIs (or their variants), our approach is able to record all storage object accesses by first- and third-party scripts. These logs are more comprehensive as compared to some prior techniques [1], which captured only the data included in HTTP requests and responses.

We then classified the object accesses based on the domain of the script which access the cookie for the first time and the domain of scripts in subsequent accesses, and categorize them as:

- Created by first-party and accessed by first-party scripts.
- Created by first-party and accessed by third-party scripts.
- Created by third-party and accessed by first-party scripts.
- Created by third-party and accessed by *same* third-party script.
- Created by third-party and accessed by *other* third-party scripts.

We further filtered the data pertaining to third-party scripts accessing the storage objects and compared these requests against some blocklists [3, 13, 41].

### 4.1 Analysis of scripts accessing storage

Table 1 shows the overall accesses of storage objects by all scripts on the web page. While 88.32% of cookie reads are done by third-party scripts, 92.53% of all cookie writes are by third-parties. Similarly, of all localstorage reads and writes, third-party scripts perform 83.76% and 98.41% of these operations, respectively; and for IndexedDB objects, 44.89% and 73.71% of all read and writes are third-party operations. In total, 89.84%, 90.98% and 72.49% of all cookies, localstorage and IndexedDB accesses **(both read and write)**, respectively, are done by third-party JS.

---

[2]https://tranco-list.eu/list/LYLP4/

| Action | # host script accesses | # 3rd-party script accesses | % 3rd-party access |
|---|---|---|---|
| **Read cookies** | 282449 | 2136907 | 88.32 |
| **Write cookies** | 102576 | 1271022 | 92.53 |
| **Read localstorage** | 43938 | 226733 | 83.76 |
| **Write localstorage** | 4154 | 258735 | 98.41 |
| **Read IndexedDB** | 167 | 136 | 44.89 |
| **Write IndexedDB** | 1798 | 5042 | 73.71 |

**Table 1: Number of different storage accesses by first- and third-party scripts on Tranco top 10K websites. The third column indicates the percentage of storage accesses done by third-party scripts as compared to total storage accesses.**

Table 2 shows the result of script access requests for cookies, localstorage and IndexedDB, which were initially created by first-party. On the other hand, Tables 3 and 4 shows detailed results of requests made to cookies and localstorage objects, which were initially created by a third-party. We omit results for IndexedDB as they were not as significant.

***Cookies:*** Out of the 10K websites, 4843 websites had scripts invoking the getCookie or setCookie APIs while the rest of the websites did not record cookie access by JS when the page was loaded during the Marionette [9] test. The results are as follows:

- 20.14% of all cookies accessed by third-party scripts are created by the host page.
- 0.54% of all the cookies created by third-party scripts are used by first-party scripts.
- 74.3% of all the cookies are shared between different third-party scripts.
- Only 4.87% of all the cookies are not accessed by third-party scripts at all.

User cookie consent is a common functionality that may require sharing cookies between scripts. For instance, OptanonConsent is a cookie used by the OneTrust CMP to store the user's consent. We observed that this cookie was being accessed by third-parties like googletagmanager.com and connect.facebook.com. A more detailed table with consent related cookies and websites in included in Table 7 in the Appendix. Some other third-parties involved in reading and modifying consent related cookies included google-analytics.com, adservice.google.com, securepubads.com, b-code.liadm.com. These domains are used by marketers and web developers to track performance, analyze user behavior, and serve targeted advertising.

***Localstorage:*** The logs captured requests from 3717 out of the 10K websites for accessing localstorage objects when the page was loaded. To summarize the accesses:

- 76% of all localstorage objects created by first-party scripts are accessed by third-party scripts.
- 0.14% of all the localstorage objects created by third-party scripts are used by first-party scripts.

- 14.66% of all the localstorage objects are shared between third-party scripts.
- 9.2% of all the localstorage objects are not accessed by third-party scripts.

***IndexedDB:*** On 913 out of the 10K websites, IndexedDB objects were accessed by JS when the page was loaded. Majority of the IndexedDB objects, i.e, 74.8%, are accessed by third-party scripts compared to only 25.2% accessed by first-party. IndexedDB objects set by first-party scripts are not accessed often by third-party scripts, as compared to cookies and localstorage objects.

***Possible violations:*** We also found a few instances where third-party scripts *not* belonging to the domain of a consent management provider modified consent-related objects. In particular, localstorage objects with keys cookieConsent* in us.diablo3.blizzard.com and blizzcon.com are modified by scripts from connect.facebook.net. Another example of such an access is the item osano_consentmanager_* used by grammy.com but modified by a script from securepubads.g.doubleclick.net.

## 5 LEAST-PRIVILEGE ACCESS

To prevent the unauthorized access of persistent storage objects by third-party scripts on a web page, we propose an approach based on the principle of least-privilege using labels.

### 5.1 Object labels

To monitor the access of storage objects in the browser, we attach labels or taints (similar to decentralized labels [37]) with cookies and other storage objects when they are stored in the browser. The labels are represented as a pair of sets of domains — $(\{r_1, r_2, ...\}, \{w_1, w_2, ...\})$ — where $\{r_1, r_2, ...\}$ lists the domains having read access on the objects and $\{w_1, w_2, ...\}$ specifies the domains whose scripts can modify the object. We additionally store the domain of the script that created these objects along with the labels. Read or write access to each object is subject to checks comparing the domain of the script accessing the object and the domains contained in the read-set or the write-set, respectively. If the domain is contained in the set, access is granted. In the default setting when the read- and write-sets of the label are empty, only the domain to which the object belongs and the domain that created the object can access that object while all other scripts are blocked access to it. We show the implications of this decision in Section 5.6.

As the owner of these objects has the best understanding of which domains to share these objects with, we rely on the developer of the scripts for setting their labels. These labels are stored across sessions (and do not reset in subsequent accesses by the user) until the objects expire or are manually deleted from the browser. Table 5 shows an example cookie jar with labels, and the respective cookie access policies.

### 5.2 Labeling cookies

To populate the labels for cookies in the browser, we introduce additional cookie attributes that can be sent by the server as part of the response (in the Set-Cookie header). These attributes specify the domains that are allowed access to read and modify the cookie. If these attributes are not specified, the cookie will be labeled with the

| Type of Access | Accessed by first-party scripts | | | | | | Accessed by third-party scripts | | | | | |
|---|---|---|---|---|---|---|---|---|---|---|---|---|
| | Get | | | Set | | | Get | | | Set | | |
| Type of storage | C | LS | IDB | C | LS | IDB | C | LS | IDB | C | LS | IDB |
| No. of websites | 332 | 868 | 41 | 675 | 405 | 173 | 628 | 326 | 67 | 195 | 127 | 215 |
| Objects accessed | 343 | 2757 | 64 | 1797 | 647 | 924 | 641 | 861 | 91 | 2989 | 179 | 2843 |
| Total accesses | 18475 | 27505 | 167 | 18840 | 3704 | 1798 | 273209 | 29140 | 136 | 71979 | 2060 | 5042 |

**Table 2: First-party and third-party scripts accessing cookies (C), localstorage (LS) and IndexedDB (IDB) objects, initially created by the host page**

| Type of Access | First Party | | Same Third Party | | Different Third Party | |
|---|---|---|---|---|---|---|
| | Get | Set | Get | Set | Get | Set |
| No. of websites | 493 | 183 | 3982 | 3802 | 2576 | 2752 |
| Objects accessed | 499 | 210 | 4747 | 29910 | 2824 | 2752 |
| Total accesses | 263974 | 83736 | 975621 | 735027 | 888077 | 464016 |

**Table 3: First-party and third-party scripts accessing cookies initially created by a third-party script**

| Type of Access | First Party | | Same Third Party | | Different Third Party | |
|---|---|---|---|---|---|---|
| | Get | Set | Get | Set | Get | Set |
| No. of websites | 177 | 47 | 2657 | 1691 | 1452 | 653 |
| Objects accessed | 518 | 54 | 16722 | 3889 | 9281 | 1541 |
| Total accesses | 16433 | 450 | 79512 | 16772 | 118081 | 13170 |

**Table 4: First-party and third-party scripts accessing localstorage objects initially created by a third-party script**

default labels having empty read- and write-sets, i.e., ({}, {}). For instance, if the domain fp.com wants to set a cookie that cmp.com has to read from, it sends the following Set-Cookie header:

`Set-Cookie: sid=123; Domain=fp.com; Reader={cmp.com}; Writer={}`

A cookie created on the client-side by any of the scripts is added to the cookie store of the host page with the script's domain having read-write access to the cookie as the owner of the cookie. For sharing these cookies with other third-party scripts, the script needs to set the proper attributes for that cookie failing which the cookie will only be accessible by the host page and the creator-script's domain. For instance, if a domain cmp.com wants to share the cookie __consent with the script from domain tkr.com, it can execute the following statement:

`document.cookie = "__consent=TRUE; Reader={tkr.com}"`

and if it needs to provide both read-write access to tkr.com, it can include the statement:

`document.cookie = "tid=567; Reader={tkr.com}; Writer={tkr.com}"`

We do not allow the owner of the cookie to be changed through JS.

## 5.3 Labeling web storage and IndexedDB objects

As other storage objects are set only through JS, we only allow scripts creating the object to specify the readers and writers of a particular object by exposing additional APIs — setReaders (key, list_of_domains) and setWriters (key, list_of_domains). Without these set, the objects are only accessible as per the default label of ({}, {}), by the script's domain other than the host page.

Note that to label all storage objects correctly via JS, the complete list of readers or writers needs to be specified, i.e., the label setting options do not append the domains to the existing sets but overwrite them with the updated values. This allows the host page to reset the labels easily without requiring an additional API to replace the existing labels.

## 5.4 Example enforcement

Recall the example in Listing 1 where the first-party script uses localstorage to store clickcount. This value can be accessed and modified by any third-party script as shown in Listing 2. The malicious script reads the value (which may also contain sensitive data) and sends it back to the server. In our proposed framework, to ensure that clickcount is only read by the analytics script, the developer could call the API setReaders(clickcount, {analytics.com}). Thus, when the script from ad_net.com tries to access clickcount, it will return an empty string as its value. Similarly, when the malicious script tries to access __consent cookie, it will receive the empty string.

Table 5 shows an example cookie jar (the first six columns) with the additional attributes. The cookie session_id can only be read by cmp.com other than the host domain. The cookie __consent can be read and modified by cmp.com as it is the creator of the cookie and only be read by tracker.com assuming that cmp.com has specified a policy allowing it to read the cookie. The third cookie tracker_id can be read and modified by all three domains as cmp.com has specified a policy allowing tracker.com to read and write.

## 5.5 Prototype implementation

We have implemented our approach on Firefox browser[3] to evaluate its efficacy. We modified the data structures and files related to cookies, localstorage and IndexedDB, and introduce new attributes for each of these storage objects. We modified (and added) around 1200 lines of code that set and check labels to control access through the different APIs. We verified that the attributes were properly set both through response headers and the JS APIs by hosting test

---

[3]We will publicly release the instrumented browser and the dataset upon acceptance.

| Name | Value | Domain | Owner | Reader label | Writer label | Script Domains | | |
|---|---|---|---|---|---|---|---|---|
| | | | | | | fp.com | cmp.com | tracker.com |
| session_id | 123 | fp.com | fp.com | {cmp.com} | {} | *RW* | *R* | - |
| __consent | TRUE | fp.com | cmp.com | {tracker.com} | {} | *RW* | *RW* | *R* |
| tracker_id | 567 | fp.com | cmp.com | {tracker.com} | {tracker.com} | *RW* | *RW* | *RW* |

**Table 5: Labels for controlling access of cookies by scripts belonging to different domains. The first three columns are part of the current cookie store while the next three columns are added by our approach to the cookie store. The last three columns indicate the privilege that the scripts from each of the three domains have for different cookies based on their labels. *R* indicates that the script has read access to the cookie in the row; similarly, *RW* and - indicate read-write and no access.**

| HTML pages, elements and scripts | Functionality | # sites |
|---|---|---|
| analytics.js | JavaScript library used to measure user activity on websites | 40 |
| activityi.html | Advertisement | 17 |
| aframe.html | Recaptcha functionality | 10 |
| dest5.html | Marketing | 5 |
| OneTrust banner | Cookie consent banner and policy of cookie usage | 11 |

**Table 6: Scripts and HTML elements missing in the manually analyzed 100 websites in the instrumented browser compared to the vanilla browser**

servers and sending requests to them through our instrumented browser.

We manually analyzed the effect of our solution for 100 websites where third-party scripts access the host-page cookies with the default policy, i.e., the readers and writers are the empty set. For this analysis, we manually saved the top 100 websites in the list of websites that contained at least one third-party script accessing cookies set by the host-page. We then compare the functionality results from the instrumented browser with the unmodified browser using the comparison tool *Meld* [7]. We discuss these results next.

### 5.6 Functionality breakage

With the default policy, in the instrumented browser, the third-party scripts are denied access to the cookies that are created or set by the host servers or the host-page scripts. We observed that the most affected functionality was the usage of cookie consent management platforms. While the number of advertisements decreased significantly, indicating better user privacy, utility (functionality) was also affected. Some recurring instances of these are listed in Table 6.

We believe that strict measures on the client-side are required for servers to add these labels as part of the Set-Cookie header sent to the client (or set them via JS). Once adopted, this approach would provide the host pages fine-grained control over how third-party scripts can access persistent storage objects.

## 6 RELATED WORK

Next, we briefly describe some of the related works in the areas of securing cookies and web-storage, and user privacy.

### 6.1 Third-party tracking and user privacy

Third-party cookies [5] have been an integral part of the user tracking, and have been used to track the online browsing behavior of users across different websites. These cookies are stored when the host page receives responses for requests to third-party domains, which unlike third-party *scripts* store cookies in the store of the third-party domain. The same third-party cookie can be used across multiple hosts to track the user activity across different host pages. Both Firefox and Chrome [2], recently introduced the idea of state partitioning [23] to prevent such stateful tracking. The main idea here is that if a.com and b.com both request content from ad.com, then a cookie from ad.com shall be saved separately for both a.com and b.com. State partitioning separates the third-party storage so that it differs for every first-party. Jueckstock et al. [31] proposed to temporarily save the third-party cookies so that it is not shared for user tracking. Our work, on the other hand, proposes a labeling-based approach for protecting first-party cookies from unauthorized scripts.

As anti-tracking mechanisms came into place, other methods took prominence. A recent work by Cassel et al. [20] discusses user tracking and browser fingerprinting techniques in mobile and desktop browsers for profiling users and as alternative ways to track the users. They show that there is a trade-off between reducing tracking and advertisement requests, and being susceptible to fingerprinting. As the defenses against third-party cookies increased, first-party storage mechanisms were used instead [24]. Munir et al. [35] also discuss the increase in the use of first-party cookies when third-party cookies are blocked. Drakonakis et al. [26] show about 5K domains which do not protect authentication cookies from JavaScript-based access while simultaneously including embedded, non-isolated, third party scripts that run in the first party's origin. Additionally, they detect 9,324 domains where sensitive user data can be accessed by such scripts (e.g., address, phone number, password).

### 6.2 Risks associated with third-party scripts

Previous research has extensively examined the prevalence of third-party scripts on websites and the associated security risks. For instance, Lauinger et al. [33] studied over 133K websites and found that 37% contained at least one script with a known vulnerability. Musch et al. [36] introduced modifications to the JavaScript environment to prevent the accidental introduction of Client-Side XSS vulnerabilities through third-party scripts. Nikiforakis et al. [38] analyzed the widespread use of third-party scripts across more than

3 million pages from the top 10,000 Alexa sites, reporting that 88.5% of popular sites incorporated at least one third-party script, often outside the main frame, and tracked the growing dependence on these scripts. Steffens et al. [40] explored the security risks by studying DOM-based client-side cross-site attacks, demonstrating how malicious or vulnerable third-party scripts can manipulate storage objects like cookies. They used dynamic taint analysis to identify behaviors leading to client-side XSS attacks, showing that third-party scripts are inherently untrustworthy. Khodayari et al. [32] complemented this by conducting a large-scale analysis of Same-Site cookie usage, examining its role in mitigating XSS-like attacks. Their work shows that privacy-preserving approaches may affect essential utilities, and optional protection mechanisms find little acceptance. These studies highlight the dynamic nature and questionable trustworthiness of third-party scripts. Hence, in our work, we focus on providing the server the means to specify what the third-party script included can do with firs-party storage.

## 6.3 Securing first-party cookies and webstorage

While we are not aware of any works targeting the security of IndexedDB objects, we discuss prior works that empirically analyze cookies and localstorage, and discuss their security.

Chen et al. [21] do a similar analysis as ours of detecting third-party scripts accessing first-party domain cookies. Their solution raises an alert on the user side if third-party scripts are tracking the user across different sites. We, however, do not focus on tracking, and instead present a generic approach to block unauthorized access of third-party scripts to all the first-party cookies. HttpOnly is used to block the same, but this restricts even the first-party scripts from accessing those cookies to provide any functionality (cookie consent example). Our approach can easily be extended to update the server about the first-party cookies changed by third-party scripts. The server can then decide whether to block them or not.

Only Belloro and Mylonas et al.[18], and Ahmed et al. [11] in their work analyze other persistent storages like indexedDB, Web SQL Database, LocalStorage and Session Storage to question the lack of user control over locally stored data. Ahmed et al. [11] used dynamic taint tracking to track the information flow from first-party scripts to third-party scripts in web browsers. Their work studies the information flows between two scripts and categorizes these flows as integrity and confidentiality flows, depending on whether the storage is written to or read by the third-party script, respectively. They found that 50% of the external (third-party domains) information flows were confidentiality flows and 30% were integrity flows. While they discuss the prevalence of possible information leakage to third-parties and the privacy implications of the same, contrary to our work, they do not discuss a solution for controlling undesired information flows.

Sanchez et al. [39] conducted an extensive analysis of cookie behavior, including the exfiltration, overwriting, and deletion of both script and HTTP cookies, across 1 million websites. In total, they collected 66.7 million cookies from 74% (738,168) of the sites they visited. Their findings show that 11% of these cookies were first-party, 47% were third-party, and 42% were classified as ghost-written. They also discovered that 13.4% of all cookies were exfiltrated, 0.19% were overwritten, and 0.08% were deleted by

scripts or via Set-Cookie headers in HTTP responses. Additionally, the study found that cross-domain cookie exfiltration occurred on 28.3% of the sites, while cookie overwriting and deletion due to collisions took place on 0.7% of the visited websites.

These studies all discuss the prevalence of first-party storage being written to by third-party scripts. However, they do not provide a solution for the same. We now discuss the different studies which propose an approach to control this behavior.

Munir et al. [35] discuss the increase of first-party tracking cookies that are being set by third-party scripts. However, they use a machine learning-based technique using data from lists such as EasyList [3], where browsers, browser extensions, or proxy servers are used to decide if certain first-party cookies may be used in user tracking by third-party scripts based on the decision given by the machine learning model. While this technique can be useful in blocking first-party tracking cookies, false positives exist affecting utility cookies such as the SSO cookies. Our analysis shows that only around 50-60% of such cookies were correctly identified and the number was even lesser in case of localstorage items; the tool heavily relies on updating these lists for proper functioning. Their work focuses on anti-tracking than actually controlling access to first-party cookies.

Bahrami et al. [14] propose a more generalized approach for managing access to first-party cookies by introducing a separate cookie jar mechanism. This jar keeps a record of the script domain responsible for setting the cookie and regulates access accordingly, ensuring that first-party cookies remain secure while maintaining normal functionality. Unlike the naive approach of outright blocking all third-party scripts from accessing first-party cookies, their solution ensures seamless performance. However, our analysis reveals instances where cross-domain interactions. Of the websites where first-party cookies were accessed (get or set request) by scripts, 45% were accessed by third-party scripts. We also observed cookies set by third-party scripts being accessed by first-party in around 5% of the cases. We also observe websites (38%) where cookies created by third-party are accessed by other third-parties. This indicates script accesses involve different domains accessing cookies or web storage created by others.

## 7 CONCLUSION AND FUTURE WORK

Third-party scripts included in sites can be a boon or a bane. While they offer richer features on the website and make the development more efficient and easier, they may introduce several threats on the host site, if not handled properly. As they are treated like any other script and have the same access to the storage objects, we propose a fine-grained approach to control this access. We introduce labels on persistent storage objects to control their access by third-party scripts. We have implemented this approach for handling cookies, localstorage and IndexedDB objects, and evaluate the enforcement on 100 websites. With proper labels in place, the instrumented browser can correctly control the access of these storage objects.

As part of future work, we want to integrate taint-tracking techniques used for information flow control with this new design to provide a complete solution for tracking data flow in the browsers.

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

## APPENDIX

| Cookie name | Website name |
|---|---|
| notion_check _cookie_consent | notion.so |
| OptanonConsent | www.eyeota.com, elisaviihde.fi, www.razer.com, www.exoclick.com, www.instructure.com, www.vonage.com, www.bodybuilding.com, www.moonpay.com, www.ledger.com, www.narrative.io, 74 more |
| gdpr_consent | www.wufoo.com |
| _pbjs_userid _consent_data | www.drugs.com, www.infoseek.co.jp, drudgereport.com , www.businessinsider.in, www.timesofisrael.com , www.cityam.com, www.rogerebert.com, www.aip.org |
| osano_consent manager_uuid | www.ada.cx, www.osano.com , www.bitcoin.com, buffalonews.com, shopping.buffalonews.com, www.linuxfoundation.org, lolesports.com, www.geotab.com, omaha.com |
| euconsent-bypass | Web.de, www.gmx.net |
| gaia_cookie_ consent-version | www.anu.edu.au |
| uncode_privacy [con-sent_types] | Lifeomic.com |
| cookiebot-consent–necessary | www.avl.com, www.aalto.fi, oscars.org, fsc.org, |
| indg-cookieConsent | www.bloombergindustry.com |
| lolg_euconsent | www.leagueofgraphs.com |
| cookie-banner-consent-accepted | www.techtudo.com.br |

**Table 7: Consent related cookies**

