# OpenReview forum: "Least Privilege Access for Persistent Storage Mechanisms in Web Browsers"
_ACM.org/TheWebConf/2025/Conference — WWW 2025 Poster_

### Official Review · Reviewer_P5jB · 2024-11-04

**Novelty:** 7
**Technical Quality:** 4

**Review:**

# Summary

This paper presents a security study on the access control of persistent storage mechanisms (cookies, localStorage, and IndexedDB) by third-party scripts in web browsers. The authors analyze the top 10,000 Tranco websites to investigate the prevalence of third-party script access to these storage objects. They propose a fine-grained access control mechanism based on the principle of least privilege, using a labeling system to specify which domains can read or write to specific storage objects. The authors implement their solution in the Firefox browser and evaluate it on 100 websites to assess functionality breakage. The study reveals that a significant portion of storage object accesses (89.84% for cookies, 90.98% for localStorage, and 72.49% for IndexedDB) are performed by third-party scripts, highlighting the need for better access control mechanisms.

# Strengths
* Addresses a critical security issue in web browsers that has not been adequately solved by existing mechanisms.
* Analyzes a large dataset of 10,000 popular websites, providing a comprehensive view of the problem's prevalence.
* Proposes a practical solution that can be implemented in real browsers.
* Implements and evaluates the proposed solution in a real browser (Firefox).
* Promises to open-source part of the dataset and framework code.

# Weaknesses
* Lack of performance analysis: The paper does not provide quantitative data on the performance impact of the proposed solution.
* Limited discussion on adoption challenges: The paper could benefit from a more in-depth analysis of potential obstacles in getting browser vendors and web developers to adopt the proposed system.

# Overall Assessment
This paper presents a valuable contribution to web security research by addressing a critical issue in the access control of persistent storage objects by third-party scripts. The empirical study provides strong evidence for the prevalence and potential risks of unrestricted access to these objects. The proposed labeling-based access control mechanism offers a promising solution that balances security improvements with maintaining necessary website functionality.

However, the paper has some limitations that somewhat reduce its impact: 1)Lack of performance analysis and limited discussion on adoption challenges leaves some important questions unanswered.  2)Lack of discussion on adoption challenges.

In conclusion, considering the strengths and limitations of this paper, I recommend a weak accept. However, If the authors commit to addressing these limitations, particularly:

* Providing a quantitative analysis of the performance impact of their solution
* Offering a more in-depth discussion on adoption challenges

I would be willing to revise my recommendation to a full accept, provided that the performance impact is within an acceptable range..

**Questions:**

* what are the performance impact of your solution.
* Why did the authors choose to evaluate their solution on only 100 websites, given the much larger initial dataset of 10,000 websites and the potential limitations this small sample size introduces in terms of representativeness and statistical significance?
* What challenges might arise when attempting to implement and scale this least privilege access control mechanism across a wider range of websites and different web browsers?

**Reviewer Confidence:**

3: The reviewer is confident but not certain that the evaluation is correct

**Scope:**

4: The work is relevant to the Web and to the track, and is of broad interest to the community

---

### Official Review · Reviewer_HB2V · 2024-11-20

**Novelty:** 3
**Technical Quality:** 3

**Review:**

Thank you for submitting to WWW 2025!

This paper investigates the security implications of third-party script access to persistent storage mechanisms in web browsers, proposing a label-based least privilege access control system.
While the topic is timely and relevant, given the increasing concerns about web privacy, several fundamental issues need to be addressed.

The label-based access control mechanism for persistent storage shows promise, but the experimental methodology and evaluation raise significant concerns. The authors' analysis of Tranco's top 10k websites needs to address more details about the dataset preparation and cleaning process. Notably, there needs to be a discussion of how CDN and DNS URLs are handled in the dataset, which could impact the results' validity. Furthermore, the paper needs to provide basic descriptive statistics about the analyzed websites, such as the average of third-party scripts per website (or per each JS), making it difficult to contextualize the findings.

A major methodological weakness lies in the classification of third-party scripts. The authors do not clearly differentiate between scripts necessary for page functionality (such as CDN-hosted libraries) and tracking scripts.
Also, the paper needs to address how the authors handled obfuscated (or packed) scripts, which third-party trackers commonly use to hide their behavior.

The data collection and analysis method needs more clarity. While the authors mention collecting information about `localStorage' and cookie access patterns, their analysis process needs to be explained more. The relationship between first-party and third-party script interactions remains ambiguous, particularly regarding access patterns and potential security implications.

While presenting some interesting findings, the evaluation section needs more explanation in several areas.
The authors need to explain the selection criteria of the sites for manual analysis.
The functionality breakage analysis, while valuable, needs to fully explore the implications of their solution in real-world scenarios, particularly for complex web applications that rely on third-party integrations.

Overall, while the paper addresses a significant security concern and presents an interesting approach, modifications are demanded to address these methodological and presentation issues. The authors should provide more rigorous justification for their methodological choices and a more precise analysis of their results.

**Questions:**

What types of information are being accessed, and what are the critical differences between first-party and third-party scripts in how this data is utilized?
Your methodology mentions using the Tranco top 10K websites, but how did you handle CDN and DNS URLs in this dataset? These should be excluded from the analysis as they represent infrastructure rather than typical websites. How many URLs were actually analyzed after accounting for these considerations? Additionally, what was your process for filtering and preparing the final dataset?

The paper needs to clearly differentiate between third-party scripts used for basic page functionality (rendering and UI components) versus those used for tracking purposes. Could you elaborate on your classification methodology?
Please provide basic statistics about your dataset, such as the average number of third-party scripts per website.
What specific data points did you collect from localStorage and cookies, and what was your analytical framework for processing this information?
Given that third-party scripts often employ obfuscation (e.g., packer or encoding to base64/32 \& Hex) techniques to hide their access patterns, how did your analysis account for and handle obfuscated code?

**Reviewer Confidence:**

3: The reviewer is confident but not certain that the evaluation is correct

**Scope:**

3: The work is somewhat relevant to the Web and to the track, and is of narrow interest to a sub-community

---

### Official Review · Reviewer_3DHE · 2024-11-25

**Novelty:** 4
**Technical Quality:** 2

**Review:**

*Summary*

The study proposes a label-based mechanism for fine-grained control of browser storage objects, limiting unauthorized third-party script access.

*Strengths*

1. Comprehensive real-world measurement of third-party scripts' access to cookies, localStorage, and IndexedDB.

2. A straightforward, label-based mechanism to define and enforce access policies.


*Weaknesses*

1. Lack of detailed description of the threat model (e.g., malicious and compromised third-party scripts).

2.Limited experimental results (Only focusing on missing scripts without broader evaluations)

*Detailed Comments*

- 4. The classification of storage accesses into five categories s insightful but not consistently reflected in the explanation. Aligning the analysis with these five categories throughout the section would improve clarity and cohesion.

- 4.1 Possible Violations: Could you provide the number of violations measured during your experiments? Quantifying these would strengthen your findings.

- 5.1 Least-Privilege Access: The terminology (e.g., "Domain" and "Owner") requires clearer definitions. For instance, does "First-Party script" imply "Domain = Owner"? Including a figure to illustrate the process would be helpful. Additionally, the threat model should be elaborated. You mentioned relying on script developers to set labels. If so, how does your method address malicious or compromised third-party scripts, as raised in your motivating example?  Further details on unauthorized access prevention are essential. One way would be to show a simple example of how to prevent an attack.

- 5.3 Examples: Could you provide specific examples of scenarios where your method is applied?

- 5.6 Experimental Results: I would appreciate more detailed results of your proposed method. Are there any performance overheads? What strict client-side measures are required? Including a list of the 100 websites tested, along with screenshots, would add clarity.
While reducing advertisements suggests improved privacy, providing quantitative evidence would substantiate this claim.
Moreover, how does your method handle situations involving cookie consent mechanisms? Is it capable of adapting client-side cookie preferences after labeling? This does not seem possible under the current implementation.

*Minor Comments*

- Tables 2, 3, and 4 require more detailed explanations. Percentages, distinctions between "get" and "set" operations, and the implications of these data points should be clarified.

- Section 6.2: The last sentence has a typo: "firt-party stroage" should be corrected to "first-party storage."

**Questions:**

- 4 How many violations were identified during your measurement study, and what do they imply?

- 5 Could you elaborate on the threat model, specifically for malicious or compromised third-party scripts? How does your method address these threats?

-  5.6 Are there performance overheads associated with your approach? If so, what are the details?

- 5.6 Does your method support dynamic changes, such as client-side cookie consents, after labeling?

- 5.6 Could you provide quantitative evidence to support claims like reduced advertisements improving user privacy?

**Reviewer Confidence:**

3: The reviewer is confident but not certain that the evaluation is correct

**Scope:**

4: The work is relevant to the Web and to the track, and is of broad interest to the community

---

### Official Review · Reviewer_haz4 · 2024-11-30

**Novelty:** 4
**Technical Quality:** 2

**Review:**

** Strength **

- Tackle an important research problem of establishing an access-control framework for browser storage mechanisms (e.g., cookies, localStorage, and sessionStorage), which is designed to restrict read/write access to these storage components, allowing read/write operations only from explicitly permitted domains.

** Weakness **

- The proposed access-control mechanism appears ineffective in achieving its designed goal.

- The measurement study in Section 4 is flawed because it is difficult to accurately attribute the sources of JS scripts in execution, which undermines the validity of the analysis.

** Comments **

Thank you for submitting your work to WWW’25. I commend the authors for addressing our community's long-standing dream(?) of establishing an access control mechanism within first-party webpages. I agree that differentiating read/write access between first-party and third-party scripts is a critical problem. I also want to see this line of research, which challenges existing web standards.

However, I noticed that two important related studies, [a] and [b], were not cited or discussed. I recommend that the authors review these works and explicitly emphasize the novelty of their proposed mechanism in comparison to these studies.

[a] BluePrint: Robust Prevention of Cross-site Scripting Attacks for Existing Browsers, Security and Privacy 2009.
[b] ConScript: Specifying and Enforcing Fine-Grained Security Policies for JavaScript in the Browser, Security and Privacy 2010.

** Measurement Study **

In Section 4, the authors conducted a measurement study to assess how many third-party and first-party scripts perform storage access initiated by first-party scripts. However, the study does not account for scripts whose origins cannot be determined. Dynamically generated JavaScript snippets executed via mechanisms like $setTimer$, $eval$, or embedded in event handlers lack identifiable domain origins, making it challenging to attribute them to specific domain sources.
Given the prevalence of dynamically generated scripts across websites, this oversight raises questions about the validity and reliability of the study’s results. Accurately attributing the origins of such scripts is itself a difficult technical challenge requiring substantial engineering effort.I recommend that the authors acknowledge this limitation explicitly in the paper and discuss how these scripts with non-deterministic origins might influence the measurement outcomes.

** Ineffective enforcement **

Based on the descriptions in Section 5.4, the proposed protection scheme requires scripts to declare their domain source in *setReaders* to access protected objects. However, the scheme does not include an enforcement mechanism to verify the authenticity of the declared domain source. If an adversary intentionally lies about their domain source, they could potentially bypass the protections by providing an allowed domain source. This lack of verification leaves the system vulnerable to deception by third-party scripts.

Moreover, third-party script providers frequently use CDNs or may change their domain names over time. These changes complicate the consistent enforcement of domain-based access control, as maintaining an up-to-date list of allowed domains becomes increasingly difficult.

As previously mentioned, dynamically generated scripts (e.g., those created with setTimer, eval, or embedded in event handlers) do not have explicit domain sources. Browsers cannot reliably infer the source domains for these scripts, further weakening the enforcement of the proposed protection scheme.

For these reasons, I do not think the proposed protection scheme is going to hold in real-world scenarios.

**Questions:**

- What is the technical novelty of the proposed system over [a,b]?

- What does the proposed system conduct access control when the third-party scripts lie about their sources by providing wrong arguments in setReader?

- How can the scripts with unknown domain sources affect the measurement results?

**Reviewer Confidence:**

4: The reviewer is certain that the evaluation is correct and very familiar with the relevant literature

**Scope:**

4: The work is relevant to the Web and to the track, and is of broad interest to the community

---

### Official Review · Reviewer_FNvU · 2024-12-01

**Novelty:** 3
**Technical Quality:** 4

**Review:**

This paper proposes a least-privilege access approach to provide fine-grained control over persistent storage objects, aiming to prevent malicious access or data alteration by third-party scripts. The authors demonstrate the risks of third-party script attacks, such as cookie tossing, and provide solid measurement results showing how persistent storage is often accessed by these scripts.
Pros
1.	The paper clearly identifies the scenarios where third-party scripts can access data objects and explains the risks of malicious attacks like cookie tossing.
2.	The experimental results are well-supported with tables, demonstrating how persistent storage is frequently accessed by third-party scripts.
Cons
1.	The core content is too brief (only 4 pages), with limited discussion on the proposed solution, its motivation, and attack measurements.
2.	The proposed solution is somewhat simplistic, and the novelty is not well-justified.
3.	The experiments are limited to 100 websites, which is small compared to the 1 million websites analyzed during the measurement phase.
4.	There are no experiments evaluating the overhead introduced by the proposed solution in terms of client-side script data access.

**Questions:**

1.	In Section 4, it seems that not all third-party accesses are malicious, and most of them may be benign. Are there any similar cases to those introduced in Section 3? Could you demonstrate or replay the attack to show how these can be a threat in practice?
2.	The labeling mechanism, while potentially introducing significant storage and timing overhead, is not evaluated in Section 5. Could you add experiments to measure these overheads, or explain why this was omitted?
3.	Do you think this mechanism is feasible for real-world applicat

**Reviewer Confidence:**

4: The reviewer is certain that the evaluation is correct and very familiar with the relevant literature

**Scope:**

4: The work is relevant to the Web and to the track, and is of broad interest to the community